# Idarucizumab in Dabigatran-Treated Patients with Acute Ischemic Stroke Receiving Thrombolytic Therapy

**DOI:** 10.3390/medicina58101355

**Published:** 2022-09-27

**Authors:** Ilga Kikule, Alise Baborikina, Iveta Haritoncenko, Guntis Karelis

**Affiliations:** 1Department of Neurology and Neurosurgery, Riga East University Hospital, LV-1038 Riga, Latvia; 2Department of Neurology and Neurosurgery, Riga Stradiņš University, LV-1007 Riga, Latvia; 3Department of Neurology, Pauls Stradiņš Clinical University Hospital, LV-1002 Riga, Latvia; 4Department of Residency, Riga Stradiņš University, LV-1007 Riga, Latvia; 5Department of Infectious Diseases, Riga Stradiņš University, LV-1007 Riga, Latvia

**Keywords:** ischemic stroke, thrombolysis, idarucizumab, dabigatran, antidote, intracerebral hematoma, functional outcome, tPA (tissue plasminogen activator)

## Abstract

*Background and Objectives*: Thrombolytic therapy with recombinant tissue-type plasminogen activator (rt-PA) is used to treat acute ischemic stroke. Dabigatran is a reversible thrombin inhibitor approved for stroke prevention in patients with nonvalvular atrial fibrillation. In such cases, thrombolytic therapy can be administered to certain patients after idarucizumab treatment. We evaluated the effectiveness of idarucizumab in dabigatran-treated patients receiving rt-PA. *Materials and Methods*: We included the data of nine idarucizumab-treated patients from the Riga East University Hospital Stroke Registry from 2018 to 2022 in our retrospective medical records analysis. We used the National Institutes of Health Stroke Scale (LV-NIHSS) score and modified Rankin scale (mRS) on admission and discharge to evaluate neurological deficit and functional outcomes. *Results*: We analyzed the data of nine patients (seven males and two females) with a mean age of 75.67 ± 8.59 years. The median door-to-needle time for all patients, including those who received idarucizumab before rt-PA, was 51 min (IQR = 43–133); the median LV-NIHSS score was 9 (IQR = 6.0–16.0) on admission and 4 (IQR = 2.5–4.0) at discharge; and the intrahospital mortality rate was 11.1% due to intracranial hemorrhage as a complication of rt-PA. *Conclusions*: Our study shows that idarucizumab as an antidote of dabigatran appears to be effective and safe in patients with acute ischemic stroke. Furthermore, the administration of idarucizumab slightly prolongs the door-to-needle time; however, the majority of cases showed clinical improvement after receiving therapy. Further randomized controlled trials should be performed to evaluate the safety and effectiveness of idarucizumab for acute ischemic stroke treatment.

## 1. Introduction

For more than 25 years, thrombolytic therapy with recombinant tissue-type plasminogen activator (rt-PA) has been widely administered in clinical practice to patients with ischemic stroke [1]. Nonvalvular atrial fibrillation is the most prevalent cause of cardioembolic stroke, which accounts for almost 30% of all stroke subtypes. Despite well-established secondary stroke prevention methods using direct anticoagulants, recurrent cerebral infarctions occur in approximately 1–3% of cases [2,3]. There is still a lack of randomized controlled trials providing information on patients with acute ischemic stroke (AIS) receiving the direct oral anticoagulant (DOAC) treatment. There is insufficient data on the use of DOAC reversal agents in these cases, as well as safety and clinical outcomes after intravenous thrombolysis (IVT) with rt-PA.

Dabigatran etexilate is a specific, reversible thrombin inhibitor approved for stroke prevention in patients with nonvalvular atrial fibrillation [4]. In such cases, thrombolytic therapy after idarucizumab administration is viable for certain patients who have used the thrombin inhibitor dabigatran for stroke prevention. This recommendation is based on the consensus of expert opinions presented in two guidelines issued in 2021 [5,6].

However, for patients taking the direct oral anticoagulant (DOAC) dabigatran, the outcome data of thrombolytic therapy with rt-PA are limited. Idarucizumab is a monoclonal antibody fragment used as a specific reversal agent for dabigatran that reverses anticoagulation within a few minutes after application without thrombotic or other side effects. It was approved by the European Medical Agency in 2015 to reverse dabigatran [7]. Recent studies have shown that patients may receive rt-PA therapy after neutralizing the effects of dabigatran, with significant clinical condition improvement [8].

## 2. Materials and Methods

We retrospectively obtained and analyzed data from the hospital stroke registry, which contains information for all hospitalized patients who fulfill the World Health Organization International Classification of Diseases 10th edition (ICD-10) criteria for an acute stroke.

### 2.1. Study Population

We collected and analyzed the data of dabigatran-treated ischemic stroke patients receiving reperfusion therapy with intravenous alteplase (rt-PA) after idarucizumab administration who were admitted to Riga East University Hospital from 2018 to 2022.

The clinical characteristics of the patients included demographic information, clinical and radiological findings, and laboratory results. We included patients who: had nonvalvular atrial fibrillation, were taking dabigatran (110 or 150 mg twice a day), were aged ≥18 years, presented with first-time or recurrent stroke with acute-onset focal neurological deficits suggestive of AIS (acute ischemic stroke), and were administered with idarucizumab before treatment with tPA. We excluded patients receiving other anticoagulants, those who had experienced hemorrhagic stroke, and those with a final diagnosis of a stroke mimic [9].

### 2.2. Measures and Definitions

We classified all acute IS subtypes using the modified TOAST criteria. The measured outcomes of the treatment were efficiency, defined as an improvement in the National Institutes of Health Stroke Scale (NIHSS) score and the modified Rankin scale (mRS) after 24 h of recanalization and safety, defined according to the occurrence of symptomatic intracerebral hemorrhage (SICH) and mortality.

### 2.3. Clinical Course and Outcome

We used the modified Rankin scale (mRS) and the Latvian version of the National Institute of Health Stroke Scale (LV-NIHSS) to measure the functional and neurological outcomes of patients [10,11].

### 2.4. Statistical Analysis

We used Microsoft Excel 2016 and IBM SPSS Statistics 24 to analyze the data of patients, the measured categorical variables (percentages and numbers), and continuous variables (median, quartiles, mean, min, max) with descriptive statistics.

## 3. Results

### 3.1. General Characteristics

Nine patients receiving idarucizumab as a dabigatran antidote before rt-PA were treated in the Riga East University Hospital Stroke Unit. All patients received a full dosage of alteplase (0.9 mg/kg). Of these patients, two (22%) were female; and seven (78%) were male. The mean age of the patients was 75.67 ± 8.59 years. The median door-to-needle time was 51 min (IQR = 43–133). 

The most common ischemic stroke localization was the left middle cerebral artery (56%), followed by the right middle cerebral artery (22%). Of the patients, 11% suffered a stroke of the left anterior cerebral artery, and another 11% suffered a combined stroke involving the left middle and posterior cerebral artery. Table 1 and Table 2 present the characteristics of patients.

### 3.2. Functional Outcome Analysis

The median LV-NIHSS score was 9 (IQR = 6.0–16.0) on admission and 4 (IQR = 2.5–4.0) at discharge, and the median mRS score on admission was 5 (IQR = 4–5). Out of all the patients, one (11.1%) had a mild neurological deficit (mRS 0–2), and eight (88.9%) were severely disabled. At the time of discharge, the median mRS score was 3 (IQR = 2–4). Four (44.4%) patients had satisfactory functional outcomes, one (11.1%) had a moderate neurological deficit, and three (33.35%) had a severe disability. The intrahospital mortality rate was 11.1% due to intracranial hemorrhage as a complication of rt-PA. Table 3 shows the functional outcomes of the patients.

## 4. Discussion

Our analysis was based on hospital stroke registry data. The mean age of our patients was 76 ± 9 years, making them similar in age or older than those included in other studies, wherein the average age was 70.0  ±  9.1 years [12] or 76 years (interquartile range 70–84) [13].

IV t-PA is a standard treatment for stroke, but guidelines currently contraindicate this therapy for patients taking DOACs. Idarucizumab administration before IVT may be effective for ischemic stroke patients taking dabigatran. IVT after dabigatran reversal with idarucizumab led to a similar rate of HT (hemorrhagic transformation), SICH (symptomatic intracranial hemorrhage), and mortality, and a similar reduction in NIHSS score compared to previous studies on non-anticoagulated patients [14]. For atrial fibrillation patients receiving direct anticoagulants who have suffered an acute ischemic stroke with a relevant neurological deficit, the most recent European Heart Rhythm Association Practical Guide recommends endovascular thrombectomy only if target vessel occlusion occurs and the procedure is deemed necessary and feasible according to present evidence [5]. None of the patients included in our analysis presented large-vessel occlusion, so they did not undergo a thrombectomy. All patients received a 5 mg bolus i/v idarucizumab dosage, as advised by the medication use recommendations and administration instructions. We observed none of the possible side effects disclosed by the drug manufacturer in the patients who received a full dosage of idarucizumab.

In our study, the median door-to-needle time was 51 min (IQR = 43–133) which is longer compared to a study of the single-centre stroke registry data of 4915 consecutive acute stroke patients who had received intravenous thrombolysis and in which the median (IQR) door-to-needle time was 42 min (32–57 min) [15]. This time deviation is most likely due to the variation in the idarucizumab administration method (two intravenous boluses of 2.5 g within 15 min vs. a single 5 g bolus). The start of thrombolysis should not be delayed by laboratory tests, but our analysis showed that the door-to-needle time of idarucizumab-treated patients was 9 min longer compared to the 22 min door-to-needle time prolongation recorded in another observational cohort study [16].

We found a clinical improvement in four (44.4%) patients, and one (11.1%) patient suffered a SICH and died. Frol et al. found that a hemorrhagic transformation (HT) was observed in 19 (7.6%) patients and a SICH in only 9 (3.6%) patients, and twenty-one patients (8.4%) died [8]. We recorded slightly higher SICH and mortality rates compared to other studies. Our results confirm the efficacy of this treatment, which was as successful as the treatment of non-anticoagulated patients. Pretreatment with dabigatran and idarucizumab at stroke onset and IVT does not confer an increased risk of hemorrhage or death. Idarucizumab is safe for patients of all regions and ethnicities and could be helpful to clinicians all around the world when tending to dabigatran-treated patients with AIS [17].

Our study had several limitations. The patient cohort was too small to perform a detailed statistical analysis. This is explained by the fact that there are very specific indications for this type of treatment. It should also be noted that only some guidelines have recommendations that support the use of idarucizumab as a reversible agent before reperfusion therapy. In other similar studies, the number of patients is relatively small as well [13,14,16]. To date, the largest patient group was explored by Kermer et al., who retrospectively analyzed the data of 120 German patients from 2016 to 2018 [18].

Larger cohorts are needed to evaluate its safety, including in relation to bleeding complications and the risk of thrombosis. However, we illustrated the potential beneficial use of idarucizumab in this situation.

## 5. Conclusions

The use of idarucizumab as an antidote for the neutralization effects of dabigatran appears to be effective and safe in certain elderly patients with acute ischemic stroke in real-world clinical trial practice. Researchers should perform further studies to evaluate the safety and effectiveness of idarucizumab use across the entire spectrum of ischemic stroke subtypes.

## Figures and Tables

**Table 1 medicina-58-01355-t001:** Demographic data of patients and door-to-needle times.

	Dabigatran-Treated Patients Receiving Idarucizumab, N = 9 (%)
Sex	
Female	2 (22%)
Male	7 (78%)
Average age (SD ^1^)	75.67 ± 8.59
Door-to-needle time, min (IQR ^2^)	51 (IQR = 43–133)

^1^ SD—standard deviation; ^2^ IQR—interquartile range.

**Table 2 medicina-58-01355-t002:** Localization of ischemic stroke.

Localization	Dabigatran-Treated Patients Receiving Idarucizumab, N = 9 (%)
Left middle cerebral artery	5 (56%)
Right middle cerebral artery	2 (22%)
Left anterior cerebral artery	1 (11%)
Left middle cerebral artery and left posterior cerebral artery	1 (11%)

**Table 3 medicina-58-01355-t003:** Functional outcomes of ischemic stroke patients.

	Dabigatran-Treated Patients Receiving Idarucizumab, N = 9
LV-NIHSS on admission (IQR)	9 (IQR = 6.0–16.0)
mRS on admission (IQR)	5 (IQR = 4–5)
0–2	1 (11.1%)
3	0 (0.0%)
4–5	8 (88.9%)
LV-NIHSS at discharge (IQR)	4 (IQR = 2.5–4.0)
mRS at discharge (IQR)	3 (IQR = 2–4)
0–2	4 (44.4%)
3	1 (11.1%)
4–5	3 (33.3%)
6	1 (11.1%)

## Data Availability

Data are available upon request due to restrictions (ethical). The data presented in this study are available on request from the corresponding author. The data are not publicly available due to localization in the hospital Stroke Registry.

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
