# Peer review of "Idarucizumab in Dabigatran-Treated Patients with Acute Ischemic Stroke Receiving Thrombolytic Therapy"

_medicina, 2022, doi:10.3390/medicina58101355_

Round 1

Reviewer 1 Report

The authors may do these corrections before publication:

The introduction is abrupt and can be improved.

The authors presented a very sample size in this retrospective study. A justification must be provided for this and for the strength of their conclusions.

The authors state “We found a significant clinical improvement in four (44.4%) patients, and one (11.1%) patient suffered a SICH and died. Data from the meta-analysis revealed HT in 19 (7.6%) patients and SICH in only 9 (3.6%) patients. Twenty-one patients (8.4%) died. We recorded slightly higher SICH and mortality rates compared to other studies.” The study though does retrospective analysis of 9 patients, the origin of these quoted observations is unclear.

Patient functional characteristics such as LV_NIHSS, pulmonary embolism, HT, SICH etc. may be tabulate per patient as the study is too small.     

Author Response

Thank you for the suggestions and editing the article

Reviewer 2 Report

This single center-observational study investigated the effectiveness of idarucizumab in dabigatran-treated patients receiving rt-PA. This study showed that the median NIHSS score was 9 (IQR = 6.0–16.0) on admission and 4 (IQR = 2.5–4.0) at discharge; and the intrahospital mortality rate was 11.1% due to intracranial hemorrhage as a complication of rt-PA. The authors suggest that idarucizumab as an antidote for dabigatran’s neutralization effects may be effective and safe in elderly patients with acute ischemic stroke. Here are my suggestions:

1.     In Abstract, conclusions are absent.

2.     What are the advantages and differences of this study from previous studies that reported the efficacy of idarucizumab? These contents should be added to the introduction, and the purpose of this study should be clearly described.

3.     A more detailed description of how and under what criteria the hospital stroke registry enrolls patients should be added. If there is a paper reported using the registry, please cite it.

4.     What efforts have been made to reduce selection bias when selecting study subjects? Were any of the patients treated with idarucizumab excluded?

5.     There is no control group to demonstrate the efficacy of idarucizumab+tPA treatment.

6.     When was the SICH and mortality for safety confirmed?

7.     What guidelines were followed for acute stroke patients after tPA?

Author Response

(The authors gave the same response as above.)

Round 2

Reviewer 2 Report

none